# Improving the Assessment and Diagnosis of Breast Lymphedema after Treatment for Breast Cancer

**DOI:** 10.3390/cancers15061758

**Published:** 2023-03-14

**Authors:** Katie Riches, Kwok-Leung Cheung, Vaughan Keeley

**Affiliations:** 1Derby Lymphedema Service, University Hospitals of Derby and Burton, Derby DE22 3NE, UK; vaughan.keeley@nhs.net; 2School of Medicine, University of Nottingham, Nottingham NG7 2UH, UK; kwok_leung.cheung@nottingham.ac.uk; 3Breast Care Unit, University Hospitals of Derby and Burton, Derby DE22 3NE, UK

**Keywords:** breast cancer, breast lymphedema, assessment, ultrasound, tissue dielectric constant, tonometry

## Abstract

**Simple Summary:**

Lymphedema affecting the breast can develop after breast cancer treatment. Currently breast lymphedema isn’t well recognised nor techniques to measure lymphedema affecting the breast as well studied. This paper explores the validity and reliability of measures which can be used to determine the presence of breast lymphedema. Women with and without breast lymphedema were included in this study to enable comparisons to be made. Improving the assessment of breast lymphedema will advance clinical practice and enable the outcome of treatment to be reported. Ultrasound and tissue dielectric constant were found to be able to reliably distinguish between edema and non edematous breast tissue. In addition certain patient characteristics and breast caner treatments were found to associated with the development of breast lymphedema.

**Abstract:**

Lymphedema can develop after treatment for breast cancer (BCRL). Lymphedema of the breast is not well studied. Currently, the main techniques used to diagnose and monitor the effectiveness of treatment are subjective clinician assessment and patient reports. Eighty-nine women who had undergone breast cancer treatment were recruited with and without breast lymphedema. Blinded clinical assessment determined the presence or absence of breast lymphedema. Measurement of skin thickness by ultrasound scanning, local tissue water by tissue dielectric constant (TDC) and tissue indentation by tonometry was recorded. Breast cancer treatment and demographic details were documented. Descriptive statistics were undertaken to compare sample characteristics, including the Chi-squared test, Odds Ratio (OR) and Relative Risks (RR) calculated. Increased body mass index (BMI), larger bra size, increased number of positive lymph nodes, axillary surgery, chemotherapy and increased Nottingham Prognostic Index (NPI) were all associated with breast lymphedema (*p* < 0.05). Ultrasound and TDC measurements were significantly higher in the lymphedema group (*p* < 0.05). Receiver Operator Characteristic (ROC) curves demonstrated that ultrasound and TDC measurements could distinguish between edematous and non-edematous breasts. Threshold levels were produced, which demonstrated good levels of sensitivity and specificity. These findings have the potential to improve the diagnosis of breast lymphedema.

## 1. Introduction

Lymphedema or chronic oedema are terms used interchangeably to describe the lymphatic system’s failure or inadequacy [1]. Oedema is often the most recognised consequence of lymphedema, but other effects include skin and tissue changes and a predisposition to infection. Lymphedema arises when there is an imbalance between capillary filtration and lymphatic drainage from the interstitial spaces, which can be due to a variety of causes [1].

Lymphedema can develop after treatment for breast cancer due to damage caused to the lymphatic drainage affecting the arm, breast, and/or chest wall. Most of the research on this topic is focused on the incidence and outcomes of breast cancer-related lymphedema (BCRL) of the arm. Lymphedema affecting the breast is not as well studied nor recognized as often in clinical practice. In some cases, the presenting signs and symptoms of breast lymphedema are misdiagnosed as inflammation occurring post-radiotherapy. Lymphedema affecting the breast can occur alongside or without arm lymphedema.

Patients often report breast heaviness, pain/tenderness, skin changes and swelling. These symptoms can be significant and impact the clothes that the patient feels comfortable wearing and the activities they are able to complete.

The assessment and diagnosis of breast lymphedema are further complicated as, currently, there are no recognised objective assessment techniques for quantifying the degree or presence of breast lymphedema. At present, it is assessed following clinical examination, patient report and, on occasion, the use of pre-and post-treatment photographs. In addition, the risk factors pertaining to breast lymphedema are not as well studied, and confounding results have been presented. This presents challenges not only in the recognition and diagnosis of the condition but in evaluating treatment outcomes.

There appears to be a paucity of studies on breast lymphedema, and this area often is omitted when lymphedema following breast cancer treatment is considered. The reasons for the lack of research in this area are not known. Breast-conserving surgery (BCS) is well-established as a surgical technique with proven effectiveness in overall breast cancer survival [2]. Breast cancer screening programmes enable earlier diagnosis of breast cancer, influencing the treatment required and enabling more women to be treated with BCS [3,4]. Therefore, the proportion of women who are at risk of mid-line lymphedema, particularly breast lymphedema, is considerable.

Breast lymphedema has been reported in several studies since as early as 1982 but does not appear to be as well recognised clinically or researched as frequently as arm lymphedema [5]. In addition, the reported prevalence varies depending on the methodology and diagnostic criteria of the study but has been reported to be as high as 75.5% [6]. Therefore, further research is required to improve the recognition and impact of breast lymphedema. In addition to the physical changes and symptoms associated with lymphedema, the literature reports that lymphedema can negatively impact a patient’s quality of life. Furthermore, this relationship is not purely linear in that quality of life does not decline as the severity of measured lymphedema increases [6,7,8,9]. Therefore, it cannot be inferred that assessing lymphedema using physical measures only can holistically assess the impact of this condition on the patient. Having breast lymphedema has been associated with poorer body image and reduced quality of life compared to those who have had breast cancer but without breast lymphedema [6].

There are only a few published studies that focus on breast lymphedema. The majority of previous work used subjective reporting to identify the presence of breast lymphedema; however, there was inconsistency in the signs and symptoms used to define it [6,10,11]. Additionally, the methodologies applied vary, including breast lymphedema diagnosed after retrospective reviewing of medical notes and prospective multiple assessment studies [5,10,11,12]. The reported risk factors associated with the development of breast lymphedema are similar to those associated with arm lymphedema. These include axillary node surgery, radiotherapy, raised body mass index (BMI) and bra size [5,10,11,12,13,14,15,16]. However, from the literature, not all risk factors were found to demonstrate statistically significant relationships, and in several studies, conflicting results were found.

Quantitative measurement techniques that are used to assess lymphedema affecting other areas have been used in studies focusing on breast lymphedema. These include ultrasound, tonometry and local tissue water measured by tissue dielectric constant (TDC) [11,12,13,14,17,18,19,20,21,22].

These have all demonstrated some potential to distinguish between edematous and non-edematous breast tissue but have limitations in their application. Additionally, diagnostic thresholds have not been proposed for each of these measurements, or the ranges suggested have not yet been validated in breast lymphedema. A diagnostic threshold for breast TDC has been produced from a cohort of women without breast lymphedema. This was calculated using the mean plus 2 standard deviations of breast TDC in a group of healthy volunteers (n = 15). A ratio of >1.4 was deemed to represent breast oedema [19]. Table 1 provides an overview of the strengths and limitations identified in using these measurements in the assessment of breast lymphedema.

The limitations identified from the available techniques to assess breast lymphedema demonstrate why no gold standard quantitative measurement technique exists.

This study’s principal aim and focus was to improve the assessment and recognition of breast lymphedema and to increase the understanding of this type of lymphedema. The overarching objective was to validate objective measures to accurately assess for the presence of breast lymphedema, which will improve clinical practice. The different objectives included:I.To consider the risk factors previously associated with the development of arm and breast lymphedema and to determine whether they apply to this study sample;II.To test and validate the objective measurement techniques identified from the literature and determine whether they can be used in the assessment of breast lymphedema;III.It was hypothesised that the measurements would differ significantly between those with and without breast lymphedema. It would be expected that they would have higher values in the group with breast lymphedema.

## 2. Materials and Methods

The design of this study enabled several objectives to be proposed with different hypotheses tested.

### 2.1. Study Design and Ethical Approval

Prior to any data collection, Research Ethics approval was obtained from the NRES Committee North West—Haydock, reference 15/NW/0608. Local approval and study sponsorship were obtained from the University of Nottingham.

The study recruited a convenience sample of women who attended the Breast Care Unit or the Lymphedema clinic. This included women with and without breast lymphedema.

### 2.2. Participants

A sample size calculation was undertaken. Applying a confidence level of 0.95, using a 2-sided interval with an expected proportion of 0.80 for sensitivity and specificity and a precision (width of confidence interval) of 0.12, a sample of 86 participants was required. This was based on 43 participants being enrolled who had clinically assessed breast oedema and 43 without.

For the measurement techniques observed to be able to distinguish between edematous and non-edematous breast tissue receiver operating characteristic (ROC) analysis was performed alongside sensitivity and specificity calculations.

The inclusion criterion for the study was that all patients were; female; >18 years of age; had undergone previous treatment for breast cancer, which included wide local excision (WLE); able to provide informed consent and had no clinical evidence of breast cancer recurrence to ensure that the presence of active disease or treatment for active disease did not influence the measurements.

### 2.3. Data Collection

All participants attended at least one study appointment at which written informed consent for study participation was obtained. At the appointment, each participant underwent blinded clinical assessment and measurement of the four breast quadrants of both breasts using tonometry, ultrasound and TDC. Breast cancer treatment data and demographic information were collected.

Although the literature review identified limitations for each measurement technique, they have all been used to determine changes in the size, tissue composition and tone of limbs to indicate the presence or absence of lymphedema. This study explored whether they can be considered valid and reliable techniques in assessing breast lymphedema. Measurement difficulties or concerns with the accuracy of each measurement were recorded to enable the application of each method to be evaluated. Measurements were undertaken in each of the four quadrants of the breast to enable comparisons to be made with the corresponding quadrant of the contralateral breast. With the exception of tonometry, repeated measurements were obtained, 2 ultrasound measurements for each quadrant and 3 TDC measurements for each quadrant. The mean of each of the measurements was calculated.

#### 2.3.1. Blinded Clinical Expert Assessment/Confirmation of Breast Oedema

The blinded clinical assessment was completed by a consultant physician with over 25 years of experience in lymphology. All participants were examined whilst lying supine. Assessment of the breast was undertaken by examining each of the four quadrants for pitting oedema individually. The “pitting” test required the clinician to press firmly for several seconds with a thumb or finger onto the examined area; once removed, if the finger/thumbprint remained, then pitting oedema was deemed present. Pitting oedema was required to be present in at least one of the four breast quadrants for that participant to be identified as having breast lymphedema. Other signs and symptoms that were assessed as part of the clinical examination included skin changes; recognised as a thickening or a peau d’orange appearance, redness or inflammation, tenderness on palpation, an increase in temperature compared to the contralateral breast and the presence of seam marks or indentations from clothes or bra. These were all assessed as present or absent.

#### 2.3.2. Measurement Using Tonometer

Prior to use, the tissue tonometer (Flinders, Australia) was calibrated using the supplied calibration plate. If it was not possible to measure all four of the breast quadrants as the tonometer could not be applied to the lower quadrants of some breasts, particularly for participants with large, ptotic breasts, this was noted. Tonometry was not repeated because the weighted probe will indent edematous tissue and, therefore, cannot be repeated in the same area until fluid in the tissues has re-accumulated.

#### 2.3.3. Measurement Using Ultrasound

Ultrasound measurements were obtained using the Sonosite Edge Ultrasound (FujiFilm Sonosite, Amsterdam, The Netherlands). A high-frequency 6–15 MHz probe was used. The participant was laid supine with the corresponding arm raised above the head. Participants were positioned this way to improve sound penetration and enable good visualisation of the breast quadrants [14]. A thick layer of ultrasound gel was applied, and the transducer was positioned perpendicular to the skin with gentle pressure applied. This was undertaken to ensure complete contact with the breast and eliminate any air pockets which could block sound waves passing through. Live images were produced by the ultrasound device from which individual frames were saved and measurements obtained. The area chosen to measure was always perpendicular, and the measurement start/end points were from the anterior echogenic border of the epidermis to the posterior echogenic border of the dermis.

#### 2.3.4. Measurement Using Tissue Dielectric Constant (TDC)

The moisture meter (Delfin Technologies, Kuopio, Finland) was used to obtain TDC measurements. An electromagnetic wave is directed into the tissue by an open-ended coaxial probe, creating an electromagnetic field in the tissue. Depending on the relative permittivity of the tissue, or dielectric constant of the tissue, the alterations in magnitude and phase of the electromagnetic wave that travels through the tissue vary. The dielectric properties of a tissue responsible for this wave shift are directly influenced by the total amount of water in a tissue. The TDC value is directly proportional to the water content in the tissue being assessed; therefore, higher values were expected in an edematous breast. Theoretically, the value obtained can range from 1, indicating no water, to 78, indicating 100% water in the measured area.

#### 2.3.5. Patient Characteristics and Breast Cancer Treatment

Patient and treatment characteristics were recorded, including; breast cancer surgery specifying the type and grade of cancer, number of lymph nodes removed and number of lymph nodes positive, adjuvant breast cancer treatment (chemotherapy, radiotherapy and hormone treatment), postoperative complications such as infections, wounds and seromas, BMI and bra size (cup and chest circumference).

### 2.4. Data Analysis

Data analysis included descriptive statistics to explore the characteristics of the whole sample group and whether there were differences between those who did and did not have breast lymphedema. This included the Chi-squared test, Odds Ratio (OR) and Relative Risks (RR).

The data from the objective measurement tools were compared and analysed using several methods. Two sample and paired t-tests or their non-parametric equivalents were undertaken to compare the different groups. Measurements from the affected breast were compared with those from the same quadrant on the contralateral breast. It was hypothesised that the measurements of both the individual quadrants and mean total breast measurements would differ in the group with clinically diagnosed breast lymphedema. It was postulated that the TDC values for the affected quadrants and the overall ratios would be higher in the lymphedema group. For the ultrasound measurements, it was expected that total cutaneous skin thickness would be thicker in the lymphedematous breast quadrants. Finally, the tonometer readings were expected to be higher in the breast quadrants with lymphedema.

Receiver Operator Characteristic (ROC) curves and the Area Under the Curve (AUC) statistics were undertaken to enable diagnostic threshold level to be produced. From these, the sensitivity and specificity calculations plus positive (PPV) and negative predictive values (NPV) and positive and negative likelihood ratios were calculated, and comparisons made of the threshold levels against the “gold standard” clinical assessment as the determinant of the presence/absence of breast lymphedema for each of the objective measurement techniques.

## 3. Results

Patients were approached over a 20-month period, and 89 women consented to participate in the study.

### 3.1. Sample Characteristics

Of the 89 participants, 40 (44.9%) were found to have breast lymphedema present. The majority of participants (29/40) had at least two quadrants of the breast that were found to have pitting oedema, with six participants having all four quadrants affected. The lower half of the breast was most commonly affected when the individual breast quadrants were compared.

The mean age of the sample was 61.1 years (standard deviation, sd = 9.6 years, range 29–80 years). The length of time from surgery to study participation ranged from 6 months to 12 years. The mean BMI was 29.25 (sd = 5.81, range 19.2–45.14). The majority of the sample was right-hand dominant (92.1%). Bra size by cup and band circumference was recorded for 81 participants. Fifty-two different bra sizes were worn by the sample, and the most common bra size was 36C. Cup size ranged from an A cup to a HH cup, and band circumference was from 34 inches to 50 inches. The entire sample had undergone a WLE, with 67 participants undergoing a sentinel node biopsy (SNB) and 23 having had an axillary node clearance (ANC). Six women had undergone both procedures, and five had not undergone any axillary procedure. The most common location of breast cancer was in the upper outer (UO) quadrant, being affected in 59.8% of the sample, followed by the lower outer quadrant (20.7%). Approximately half of the group had a grade II tumour, and one-third had a grade III tumour. Twenty-two (26.2%) were found to have lymphatic or vascular invasion (LVI) present on histopathology. The Nottingham Prognostic Index (NPI) ranged from 2.06–6.8, with a mean NPI of 3.98 (sd 1.102). The number of lymph nodes (LN) removed ranged from 0–38. The mean number of LN removed was 6.17 (sd 8.442), and the median number of nodes removed was two (interquartile range, IQR, 1–10). The number of LN removed, and the presence of metastatic deposits varied depending on the extent of the procedure(s) to the axilla. Adjuvant treatments included chemotherapy (40/84, 44.9%), radiotherapy (88/89, 98.9%) and hormone therapy such as Tamoxifen, Anastrazole and Letrozole (65/84, 73%). The chemotherapy regimens varied depending on the length of time between cancer treatment and study participation. Details of radiotherapy treatment were recorded for 87/88 participants who received it. The entire group received radiotherapy to the breast, the most common dose being 40 Gray (Gy) (70%).

### Breast Cancer Treatment and Breast Lymphedema

Comparison of the type of axillary surgery or at least one positive lymph node in the presence of breast lymphedema was undertaken using the X^2^ test. More participants were observed than expected with breast lymphedema in the ANC or positive lymph node group; this difference was statistically significant in both cases (see Table 2).

Further calculation of the relative risk demonstrates that patients who underwent an ANC or had at least one positive LN were almost twice as likely to develop breast lymphedema as those with an SNB or negative lymph nodes. These groups were not mutually exclusive. Although most participants with lymph node-positive disease had undergone an ANC, some participants with lymph node-positive breast cancer had only undergone an SNB.

The entire sample except one patient received radiotherapy; therefore, no comparisons could be made between those who had or did not have radiotherapy. Table 3 displays the number of participants who received chemotherapy and hormone therapy. The X^2^ test was not significant when hormone therapy was considered. However, more participants in the breast lymphedema group had received chemotherapy (*p* = 0.031, RR = 1.657).

There was no significant difference between the age of the participant and the presence or absence of breast lymphedema (*t*-test, *p* = 0.375).

Other characteristics did demonstrate statistically significant differences between those with and without breast lymphedema. The mean values for weight, BMI and NPI were higher in the lymphedema group (*p* < 0.001, 0.001 and 0.04, respectively).

Median chest circumference was higher in the lymphedema group at 40 inches, compared to 36 in the non-lymphedema group (*p* < 0.001), see Figure 1.

Comparing bra cup size, the distribution varied between the two groups; the lymphedema group wore larger cup-sized bras (Figure 2). The majority of the group without breast lymphedema wore a C cup or smaller (63.8%); however, this cup size was less frequently worn in the lymphedema group (23.5%). The most commonly worn bra cup in the lymphedema group was a DD (20.6%). The Mann-Whitney U test confirmed a statistically significant difference between these groups (*p* < 0.001).

### 3.2. Ability to Distinguish Edematous and Non-Edematous Breast Quadrants Using Tonometry

Comparing the affected and unaffected measurements obtained by the tonometer, *p* values were >0.05 for each of the four breast quadrants. This result supports that the null hypothesis is accepted and that there is no significant difference in the tonometer measurements for any of the four quadrants A comparison of the median values demonstrated that these are similar in both groups.

### 3.3. Ability to Distinguish Edematous and Non-Edematous Breast Tissue by Ultrasound

Using the paired sample *t*-test, mean skin thickness ultrasound measurements were significantly higher in the affected breast quadrant compared to the contralateral (unaffected) breast quadrant (all *p* < 0.05) for each of the four quadrants (Table 4). In each of the four quadrants, the mean skin thickness in the affected group was approximately double the measurements of the corresponding unaffected quadrant. The inner quadrants were thicker in both the affected and unaffected groups compared to the outer quadrants. As there were fewer participants with lymphedema in the upper quadrants, the numbers in these groups were smaller.

### 3.4. Ability to Distinguish Edematous and Non-Edematous Breast Quadrants Using TDC

Applying the paired samples *t*-test, mean TDC readings were significantly higher in each of the affected breast quadrants compared to the unaffected breast quadrants. These were comparable with the ultrasound measurements. The inner quadrants had higher TDC readings, but the difference between these and the outer quadrants was by a few units only (Table 5).

When TDC is measured commonly, the raw data is not used, but a comparison is made between ratios comparing affected and unaffected area(s). Using the Independent samples *t*-test, the mean ratio was significantly higher in the group with breast lymphedema than in the non-lymphedema group (*p* < 0.001) (Table 6). The measurements from the four breast quadrants were added together, and a ratio of the affected to the unaffected breast was produced (Table 7).

### 3.5. Receiver Operating Characteristic (ROC) Curves

#### 3.5.1. ROC Analysis for Tissue Dielectric Constant for the Whole Breast

The AUC statistic for TDC is 0.901, standard error = 0.032 and produced a 95% confidence interval of 0.839–0.964 (Figure 3). Analysis of the ROC curve identified a TDC threshold using a ratio of 1.34, producing a sensitivity of 87.5% and specificity of 79.6% (Table 8).

The high PPV and NPV identified that approximately 78% and 87% of participants with or without breast lymphedema were correctly identified using the TDC threshold of 1.34. The TDC threshold ratio of >1.34, the positive likelihood ratio (+LR) of 4.29 and the negative likelihood ratio (-LR) of 0.157 indicate that these tests may improve the assessment of breast lymphedema.

#### 3.5.2. ROC Analysis for Ultrasound Measurement

The AUC, associated *p* values, sensitivity and specificity results support the use of skin thickness measurement of the breast quadrants by ultrasound scanning in the identification of breast lymphedema (Table 9). The PPV identified that approximately 44–58% of the sample with a positive test did have breast lymphedema (Table 10). However, the NPV values are much higher, 87–99% indicating that participants with a “normal” ultrasound measurement did not have breast lymphedema. Comparable to TDC data, the positive and negative likelihood ratios for the USS thresholds again indicate that these tests add value to the assessment of breast lymphedema.

## 4. Discussion

This study had several aims, all associated with improving the diagnosis and recognition of breast lymphedema after breast cancer. It appears that some of the risk factors associated with the development of lymphedema affecting the arm are also risk factors for the development of breast lymphedema. In addition, the relationship between increased breast size and the development of breast lymphedema was established in this study. The measurement of skin thickness by ultrasound scanning and skin water measurement by tissue dielectric constant were both valid methods for assessing breast lymphedema. ROC analysis produced threshold values that could be applied in practice to distinguish between edematous and non-edematous breast tissue. In this study, the tissue tonometer was not found to be reliable in distinguishing between the edematous and non-edematous breast tissue and, therefore, not recommended as an assessment tool for breast lymphedema.

The analysis identified that the treated breast skin was thicker by ultrasound measurement and had a higher TDC reading than the non-treated breast, even in participants who did not have breast lymphedema.

Repeated assessment confirmed the reliability of the ultrasound and TDC measurements. Repeated measurements did not differ significantly when comparing the initial and second assessment measurements.

The characteristic of the study sample in relation to age, breast cancer diagnosis and treatment are comparable to the UK breast cancer population, which supports the findings being applied to clinical practice [24,25,26]. Breast lymphedema is not well studied but is a growing topic. Participants recruited in this study had undergone breast cancer surgery up to 12 years earlier, recognising breast lymphedema as an ongoing problem for some women.

### 4.1. Risk Factors

Several risk factors pertaining to the development of breast lymphedema were identified, including lymph node-positive breast cancer, ANC, NPI, BMI and bra size. This is valuable information which will support the education of patients undergoing breast cancer treatment. It has been suggested that breast lymphedema should be considered and the risks of developing this explained to patients to ensure informed treatment decisions are made [27]. In addition, identifying those at higher risk could be used to determine who might benefit from monitoring post-operatively.

Having lymph node-positive disease and ANC demonstrated an increased risk of breast lymphedema than those with lymph node-negative disease or requiring an SNB. This is unsurprising as the participants who underwent an ANC would have done so due to having had a malignant axillary lymph node identified as part of their breast cancer diagnosis. Previously, in a large prospective study of breast oedema, ANC and adjuvant chemotherapy were identified as associated risk factors at time points up to 18 months post-radiotherapy [27]. This finding may also be related to the risk factors; receipt of chemotherapy and high NPI, reflecting that breast lymphedema may develop because of more advanced breast cancer and the intensive treatment it requires. Future research focusing on confounding variables and multicollinearity would provide information on whether these are individual risk factors or interrelated.

In this study, larger chest circumference and larger bra cup size were associated with the presence of breast lymphedema. In the general population, bra or bust size has been recognised to be increasing as the body size of the population increases [28]. The average UK bra size was reported to have increased from 34B in 2008 to a 36DD in 2019 [28].

Over two-thirds of participants received surgery to remove a tumour from the upper breast quadrants; however, only one participant had oedema present in the upper quadrant only. For those who had oedema in one or two of the four breast quadrants, it was in the lower quadrants. This raises the question of whether it is breast cancer treatment, including surgery and radiotherapy, which impairs the lymphatic drainage from the lower half of the breast, which causes oedema. A similar presentation of lymphedema is observed when hand and forearm swelling develops after breast cancer treatment, which is focused away from the operated area of the breast and axilla. Questions regarding the lymphatic drainage pathways of the arm and breast following axillary surgery have recently been raised, specifically whether, in some cases, they are able to regenerate themselves, the surgical breaks filled and the drainage pathways repaired [29]. In a recent study, imaging of the arm and breast lymphatics using indocyanine green (ICG) fluorescence lymphography after ANC treatment for breast cancer demonstrated several different variations in lymphatic flow. In addition to regenerated lymphatic pathways seen, lymphatic drainage appeared to cross the midline and drain into the contralateral axilla [29]. An alternative consideration is whether increased breast size creates a venous hypertension effect on the breast resulting in more fluid for the lymph system to drain. This may be contributed to by the gravitational effect of larger, ptotic breasts. Such a phenomenon is recognised in lymphedema associated with significant obesity that affects the abdominal apron or other areas and the development of massive localised lymphedema [30]. In such cases, increased capillary filtration with overloaded regional lymphatics results in oedema development [30]. These symptoms are also associated with the presence of chronic oedema of the legs due to venous hypertension.

### 4.2. Assessment of Breast Edema

In this study, the tissue tonometer could not distinguish edematous and non-edematous tissue, the median readings obtained were similar, and the analysis did not reach significant levels. The defining characteristic for determining oedema in any breast quadrant was the presence of pitting oedema during the clinical assessment. This mimics the technique of the tonometer. It was, therefore, surprising that the tonometer did not record higher readings in the edematous breast quadrants.

In addition, readings could not be obtained for all the study participants as the tonometer could not be positioned correctly to enable the measurement to be undertaken; this was more common when measuring the lower breast quadrants.

Measurement of dermal thickness by high-frequency ultrasound scanning demonstrated this technique to be valid and reliable in assessing breast lymphedema. This study has shown that ultrasound measurements can be used to distinguish edematous and non-edematous breast tissue, and reproducible measurements were obtained at repeated assessments. The procedure was well tolerated, and measurements were able to be performed on all of the participants in this study.

Similar to other studies, the measurements obtained in this study differed depending on the quadrant or part of the breast being measured. The ultrasound measurement points, and the number of measurements obtained in this study vary from other studies, including a recently published study [31]. The medial (inner) aspect(s) of the breast in all of the studies, including this study, were thicker than the lateral (outer) aspect(s). This relationship was consistent for the affected and unaffected breasts. In this current study, the highest measurement for the edematous breast was in the upper inner quadrant (UIQ), and this varied from the literature, which identified the lower breast quadrants as the thickest. The upper outer quadrant (UOQ) was the thinnest in this and all of the studies reported in the literature for both the affected and unaffected breasts. In addition, the TDC readings were lowest in this quadrant. This was an unexpected result as if the lymphatic drainage of the breast is through the axillary lymph nodes, it would be postulated that there would be more fluid draining through this quadrant resulting in higher USS and TDC measurements. Additional research using live imaging techniques to assess lymphatic flow, such as lymphoscintigraphy or ICG lymphography, would be useful to study this finding further.

In this study, there were only 11 participants who had oedema in the upper inner breast quadrant; therefore, any outliers in this group would potentially have increased the mean of the measurements obtained. However, the confidence interval and standard deviation produced in this subgroup are similar to those produced for the other subgroups for both USS and TDC measurements.

However, the USS measurements proposed in this current study differ from the thresholds suggested in a recently published study [31]. In that study, the USS cut-offs were 1.6 mm (outer quadrants) and 2.0 mm (inner quadrants). The thresholds proposed in this study are higher for each of the four quadrants. This difference may result from the measurement points not being identical in the two studies. In this study, the measurements were obtained in the middle of the breast quadrants and on the boundary between quadrants in the other study. The high negative predictive values from this current study demonstrated that using these measurements as a threshold for diagnosis would result in few false negative classifications. Therefore, the clinicians would be able to provide reassurance or confidence in diagnosing breast lymphedema. Normal breast skin thickness varies between 1 and 2 mm, with a mean of 1.7 mm [32]. The thresholds suggested in this study all exceed 2 mm. Therefore, a higher diagnostic threshold, such as that proposed in this current study, might be preferred to reduce overdiagnosis and false positives.

The TDC ratio threshold proposed in this study (1.34) is at the midpoint between the two previously proposed ratios, 1.4 for breast oedema and the initial TDC threshold of 1.26 for determining the presence of forearm lymphedema [20,23]. Using either the 1.4 or 1.34 ratio threshold increases the specificity and PPV but reduces the sensitivity and NPV. None of the TDC thresholds correctly diagnosed the entire sample, which was to be expected. If an assessment technique is used to screen patients and identify those who may have the condition in question for further assessment or to reassure those at risk of the condition that they do not have it, then a threshold may be selected due to the desired sensitivity, specificity, PPV or NPV.

The price, portability and simple training required to use the moisture meter, compared to obtaining ultrasound images, would identify this as a technique that could be used in the screening and preliminary assessment of breast lymphedema. Therefore, a lower threshold, such as the 1.34 proposed in this study, could be used as the threshold for breast lymphedema screening when referral to a lymphedema service is recommended for a more detailed assessment to be undertaken.

### 4.3. Limitations of the Study

Limitations of this study include that a single clinician undertook the clinical assessment. It had been hoped that an additional physician would also be present to complete a second assessment, enabling inter-rater reliability to be examined. On reflection, we could have obtained photographs of the participants’ breasts which could have been reviewed by other clinicians in the team. In addition, the temperature of the breasts could have been recorded, which would have provided a physical measurement and strengthened the assessment. The identification of raised breast temperature was a subjective assessment following palpation of each breast.

The analysis identified that some patients appear more likely to develop breast lymphedema due to individual characteristics, breast cancer treatment and cancer histology. This study was not powered to undertake multivariate analysis; further research with a larger sample is needed. This information would enable clinicians to discuss individual risks and identify which patients might benefit from additional monitoring.

## 5. Conclusions

The findings from this study add to the existing knowledge base and have a strong clinical application that can be applied to improve the assessment and diagnosis of breast lymphedema. Identifying potential risk factors provides information that can be shared with patients before breast cancer treatment to educate them on the potential risk of lymphedema development. Breast lymphedema appears more common in patients with larger breasts and/or more advanced breast cancer.

In addition, the measurement thresholds proposed for USS and TDC could be applied in clinical practice to aid the diagnosis of breast lymphedema. Further research using these techniques would enable the treatments provided for breast lymphedema to be evaluated.

## Figures and Tables

**Figure 1 cancers-15-01758-f001:**
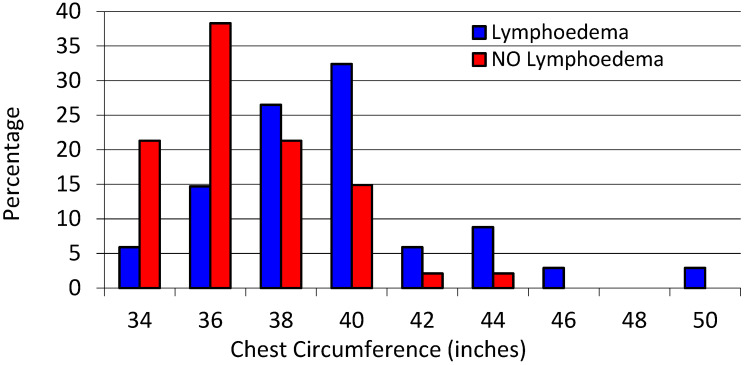
Comparison of Chest circumference.

**Figure 2 cancers-15-01758-f002:**
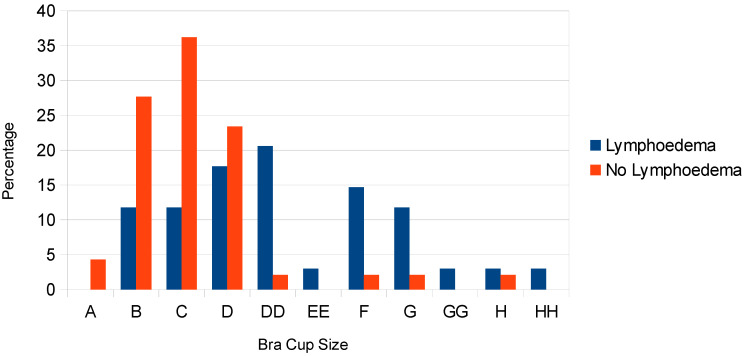
Comparison of Bra Cup Size.

**Figure 3 cancers-15-01758-f003:**
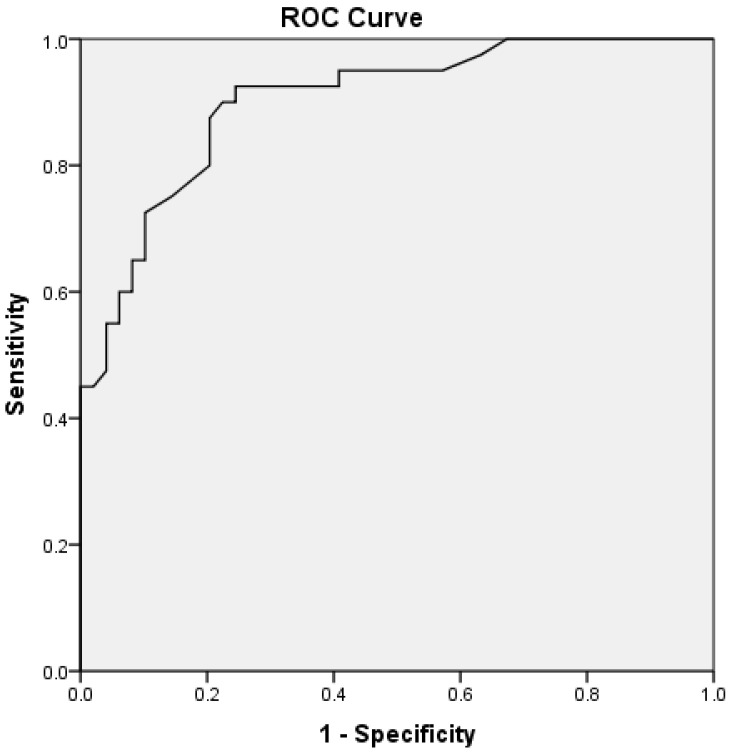
ROC curve for TDC ratio.

**Table 1 cancers-15-01758-t001:** Advantages, disadvantages and limitations of the measurement methods reviewed.

Method	Application	Advantages in Assessment	Limitations in Assessment
Ultrasound [10,13,16,21,22]	Longitudinal follow-up and assessment of breast oedema. Recognition of changes in breast tissue prior to RT and resolution of oedema over time. No pre-surgery USS measurements were obtained, but pre-and post-RT, up to 24 months.	Can be used to compare to unaffected breasts. Can measure and assess each breast quadrant separately. Most commonly studied assessment technique.	Difficulty distinguishing between the layers of the skin. Therefore total cutaneous thickness was recorded. Mean coefficient of variance = 13.9%. Assessment is not always correlated with clinical examination.
Tonometer [17]	Reproducibility study undertaken. Used as an outcome measure in the identification of capsule formation post-breast reconstruction. Not used in a longitudinal study for assessing breast oedema.	Successfully tested for reproducibility in the assessment of breast oedema. Measures local tissue changes/differences with contralateral points.	Not studied the difference between affected/unaffected breast or breast quadrant measurements. Breast shape changes depending on position and natural differences in shape and tone of the breast.
Tissue Dielectric Constant (TDC) [18,19,20,23]	Publications to date on the assessment of local tissue changes of the limbs.	Has been used to measure local fluid changes/differences; therefore, it could measure individual breast quadrants.	Normal or reference values were published for specific measurement points on the arm, and the ratio of 1.26 was identified as indicative of the presence of oedema. More recently, a ratio of >1.4 has been proposed as indicative of the presence of breast oedema.

**Table 2 cancers-15-01758-t002:** Chi-Squared test comparing axillary surgery and the presence of positive lymph nodes. (Obs = observed (proportion).

	Positive LN		Negative LN		ANC		SNB	
	Obs		Obs		Obs		Obs	
Breast Lymphedema	19 (0.66)		20 (0.37)		16 (0.7)		23 (0.38)	
No Lymphedema	10 (0.34)		34 (0.63)		7 (0,3)		38 (0.62)	
X^2^ statistic	6.144				6.816			
*p*	0.013				0.009			
Relative Risk (95% CI)	1.769 (1.143–2.737)				1.845 (1.211–2.810)			

Abbreviations: LN = lymph node, ANC = axillary node clearance, SNB = sentinel node biopsy.

**Table 3 cancers-15-01758-t003:** Chi-squared test comparing receipt of chemotherapy and hormone therapy.

	Chemo		No Chemo		Hormones		No Hormones	
	Obs		Obs		Obs		Obs	
Breast Lymphedema	23 (0.58)		17 (0.35)		33 (0.51)		7 (0.29)	
No Lymphedema	17 (0.42)		32 (0.65)		32 (0.49)		17 (0.71)	
X^2^ statistic	4.629				3.306			
*p*	0.031				0.069			
Relative Risk (95% CI)	1.657 (1.038–2.645)				1.741 (0.893–3.394)			

**Table 4 cancers-15-01758-t004:** Comparison of the Tonometer Measurements of the Individual Breast Quadrants using the Wilcoxon Signed Rank test.

Breast Quadrant	Breast Lymphedema (LE) or No Lymphedema (No LE)	N	Median (IQR)	*p* Value
Lower Outer	LE	29	0.91 (0.88–0.95)	0.443
	No LE	28	0.93 (0.90–0.94)	
Lower Inner	LE	26	0.925 (0.90–0.94)	0.074
	No LE	26	0.94 (0.92–0.94)	
Upper Outer	LE	11	0.92 (0.88–0.95)	0.575
	No LE	11	0.90 (0.88–0.94)	
Upper Inner	LE	11	0.95 (0.92–7.8)	0.398
	No LE	11	0.94 (0.94–0.96)	

**Table 5 cancers-15-01758-t005:** Paired *t*-test comparing skin thickness measurement by ultrasound scanning of the individual breast quadrants.

Quadrant	Breast Lymphedema (LE) or No Lymphedema (No LE)	N	Mean (mm)	Standard Deviation	Mean Difference (Confidence Interval)	Standard Error of Difference	*p* Value
Upper Outer	LE	12	3.20	0.885	1.50 (0.83–2.17)	0.30	<0.001
	No LE	12	1.70	0.400			
Lower Outer	LE	31	3.73	1.527	2.12 (1.61–2.63)	0.25	<0.001
	No LE	31	1.62	0.340			
Lower Inner	LE	29	4.06	1.445	2.21 (1.69–2.73)	0.25	<0.001
	No LE	29	1.847	0.481			
Upper Inner	LE	11	4.19	0.954	2.205 (1.62–2.80)	0.26	<0.001
	No LE	11	1.98	0.237			

**Table 6 cancers-15-01758-t006:** Tissue Dielectric Constant of the individual breast quadrants, comparing the edematous to contralateral quadrants.

Quadrant	Lymphedema (LE) or No Lymphedema (No LE)	N	Mean	Standard Deviation	Mean Difference (Confidence Interval)	Standard Error of Difference	*p* Value
Upper outer	LE	12	40.21	8.776	15.173 (10.02–20.33)	2.34	<0.001
	No LE	12	25.04	4.516			
Lower outer	LE	31	45.09	11.425	19.743 (15.25–24.24)	2.20	<0.001
	No LE	31	25.34	5.817			
Lower inner	LE	29	47.37	10.517	17.699 (12.71–22.69)	2.44	<0.001
	No LE	29	29.67	7.055			
Upper inner	LE	11	49.51	11.310	2.426 (17.76–28.58)	2.42	<0.001
	No LE	11	26.34	4.718			

**Table 7 cancers-15-01758-t007:** Comparison of the Tissue Dielectric Constant Ratios (Affected Breast: Unaffected Breast).

	N	Mean	Standard Deviation	Mean Difference (Confidence Interval)	Standard Error of Difference	*p* Value
Breast Lymphedema	40	1.624	0.306	0.450 (0.340–0.561)	0.055	<0.001
No Lymphedema	49	1.174	0.188			

**Table 8 cancers-15-01758-t008:** Sensitivity and specificity of TDC ratio at 1.34 threshold range.

	TDC Ratio > 1.34	TDC Ratio < 1.34
Breast Lymphedema	35	5
No Lymphedema	10	39
Sensitivity (Confidence Interval)	87.5% (77.6–97.7%)	
Specificity	79.6% (68.3–90.9%)	

**Table 9 cancers-15-01758-t009:** Ultrasound AUC statistics plus proposed reference ranges, sensitivity and specificity.

	AUC	Standard Error	*p*	95% CI	Proposed Threshold (mm)	Sensitivity	Specificity
LOQ	0.898	0.035	<0.001	0.83–0.966	≥2.3	83.9%	87.1%
LIQ	0.898	0.030	<0.001	0.839–0.958	≥2.6	79.3%	83.9%
UOQ	0.946	0.026	<0.001	0.896–0.997	≥2.5	91.7%	91.6%
UIQ	0.972	0.016	<0.001	0.94–1.00	≥3.0	90.9%	93.4%

Abbreviations: LOQ = lower outer quadrant, LIQ = lower inner quadrant, UOQ = upper outer quadrant, UIQ = upper inner quadrant.

**Table 10 cancers-15-01758-t010:** Ultrasound Positive and Negative Predictive Values and Likelihood Ratios.

	PPV	NPV	+LR	−LR
LOQ	57.8%	87.1%	6.504	0.185
LIQ	48.9%	95.4%	4.925	0.247
UOQ	44.0%	99.3%	10.917	0.091
UIQ	47.6%	99.4%	13.773	0.097

## Data Availability

Further information on the data collected can be sought from the authors (K.R.).

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
