# Peer review of "Improving the Assessment and Diagnosis of Breast Lymphedema after Treatment for Breast Cancer"

_cancers, 2023, doi:10.3390/cancers15061758_

Round 1

Reviewer 1 Report

This study focus the attention on testing and validating objective
measurement techniques in the assessment of breast lymphedema and risk
factors.
In my opinion this study is interesting and well conducted.
Radiotheraphy plays an important role in breast lymphedema. In this
study the entire sample except one patient received rediotherapy (most
common dose 40 Gray). It would be interesting to investigate
radiotheraphy role by comparing different doses.

Author Response

Thank you for your review and comments. I did collect the radiotherapy site(s) and dosage for those patients who had received their breast cancer treatment locally. All of the 88 patients who underwent radiotherapy included treatment to the breast and at least 40Gy.

Reviewer 2 Report

Dear author

Thank you for the submission of your article to our journal. I’ve just read your paper and found many problems listed below.

Major points

Lymphedema of the upper extremities significantly reduces the patient's QOL, so it is of great significance to correctly evaluate and treat lymphedema. On the other hand, it is necessary for the authors to first describe in detail the clinical problems associated with breast lymphedema, which is the subject of this study. The authors need to show the direction of how this clinical problems can be prevented or alleviated in this study.

Minor points

Why do you list Table 7 before Table 4?

Where in the text are Table 2, 9, and 10 mentioned?

Table 1&Lines 386 and 394

What’s the meaning of USS?

Fig.1 

NO Lymphoedema→No lymphedema

Fig.2

No lymohoedema→No lymphedema

Line 250

these are similar in both groupsthese were similar in both groups 

Line 259

these groups are smallerthese groups were smaller 

Table 8

When the case with a TDC ratio of 1.34 was observed, did you categorize it in which group?

Line 207

Where is table 13?

Line 315

education of patients of patients  education of patients

Line 323

The expression "a malignant axillary lymph node” is strange.

Line 357

What’s the meaning of “IThese symptoms”?

Author Response

Major points

Lymphedema of the upper extremities significantly reduces the patient's QOL, so it is of great significance to correctly evaluate and treat lymphedema. On the other hand, it is necessary for the authors to first describe in detail the clinical problems associated with breast lymphedema, which is the subject of this study. The authors need to show the direction of how this clinical problems can be prevented or alleviated in this study.

R: Thank you for taking the time to review my paper and for the comments made. I have now been through each of these and amended the paper accordingly. 

Minor points

Why do you list Table 7 before Table 4?

R: Sorry this was a mistake and has been corrected.

Where in the text are Table 2, 9, and 10 mentioned?

R: All tables and acknowledgement in the manuscript checked and now included.

Table 1&Lines 386 and 394

What’s the meaning of USS?

R: Ultrasound -this explanation of the abbreviation has been included

Fig.1 

NO Lymphoedema→No lymphedema

R: Amended

Fig.2

No lymohoedema→No lymphedema

R: I cannot amend this as the figure has become a picture and I cannot change the legend

Line 250

these are similar in both groups→these were similar in both groups 

R: changed

Line 259

these groups are smaller→these groups were smaller 

R: changed

Table 8

When the case with a TDC ratio of 1.34 was observed, did you categorize it in which group?

R: There were no ratios of 1.34 obtained but this has been corrected to read greater or equal to 1.34

Line 207

Where is table 13?

R: corrected

Line 315

education of patients of patients → education of patients

R: corrected

Line 323

The expression "a malignant axillary lymph node” is strange.

R: Changed to positive lymph node

Line 357

What’s the meaning of “IThese symptoms”?

R: Changed to "these"

Round 2

Reviewer 2 Report

Dear author

Thank you for the re-submission of your article to our journal. I found that a few corrections were found, but many of the problems I pointed out in the first review were not corrected in your revised manuscript.

Author Response

Dear Reviewer,

Thank you for taking the time to read and review my paper. I do apologise if I haven't considered all of the changes you made. I think / hope that I have read through each of them plus those of the other reviewer and the journal editor and amended the paper for each point. I had not used tracked changes when I first revised the paper but have now been through and highlighted each changed part.

As part of my study I looked specifically at quality of life and validated a QoL tool for breast lymphoedema. This work is a separate paper. There was so much in this paper already looking at the objective assessment of breast cancer it felt that the QoL study needed to be written and hopefully published separately. 

Thank you again,

Katie

Round 3

Reviewer 2 Report

Dear author

Thank you for the re-submission of your article to our journal. I found that a few corrections were found, but many of the problems I pointed out in the first review were not corrected in your revised manuscript.

Author Response

Dear Reviewer,

Thank you for taking the time to read again my paper and the changes made. I apologise that you feel I haven't taken on board or amended the paper in line with your comments. I hope I have now addressed them.

Following your first review I amended each of the comments/suggestions. For the second and third reviews I have looked at the suggestions about describing the clinical problem of breast lymphoedema and how this study could help prevent / alleviate this problem. I hope now that I have clearly described how breast lymphoedema affects those who develop it, that it is not as well recognised and that there isn't the techniques to accurately / consistently diagnose it. 

I am hopeful that the findings of this study help raise the awareness of breast lymphoedema, highlight those patients who appear to have a higher risk of developing it. Additionally TDC and USS can be used to provide reliable measurements to help aid diagnosis and evaluate treatment.

Within my study I developed and tested a QoL tool. The results of this will be written up and hopefully published, building further on the assessment and impact of breast lymphoedema.

I hope now that I have made sufficient changes to the paper.

Thank you again for your time and comments.

Round 4

Reviewer 2 Report

Dear author

Thank you for the re-submission of your article to our journal. I’ve just read your paper and consider the changes to peer review points to be acceptable, though not sufficient. I, therefore, accept this revised paper.

Author Response

Thank you for your time and your comments.

Katie